# BREADTH-FIRST EXPLORATION IN GRID-BASED REINFORCEMENT LEARNING

## ABSTRACT

Recently, graph-based planners have gained significant attention for goal-conditioned reinforcement learning (RL), where they construct a graph that represents confident transitions between "subgoals" as edges and run shortest path algorithms to exploit the confident edges. Such a graph construction consists of only achieved subgoals while recording unattained ones in history is also crucial. Indeed, it often wastes an excessive number of attempts to unattainable subgoals. To alleviate this issue, we propose a graph construction method that efficiently manages all the achieved and unattained subgoals on a grid graph adaptively discretizing the goal space. This enables a breadth-first exploration strategy, grounded in local adaptive grid refinement, that prioritizes broad probing of subgoals on a coarse grid over meticulous one on a dense grid. We empirically verify the effectiveness of our method through extensive experiments.

## 1 INTRODUCTION

Many real-world sequential decision-making problems can be framed as the task of targeting a given goal, e.g., the navigation of walking robots (Schaul et al., 2015; Nachum et al., 2018) and the manipulation of objects using robotic arms (Andrychowicz et al., 2017). Goal-conditioned reinforcement learning (RL) aims to solve these problems using a goal-conditioned policy designed to maximize the return with respect to the target goal. This offers a versatile policy for a variety of distinct problems, described by corresponding goals, whereas other RL frameworks often require separate policies for different tasks. Besides the inherent versatility, it also enables the hierarchical RL (Zhang et al., 2020) which decomposes a daunting long-horizon goal into a series of manageable short-horizon subgoals so that the agent can exploit more confident and learnable transitions between subgoals than the direct transition to the ultimate goal.

Recent advancements in RL have witnessed the emergence of graph-based planners as potent tools for this subgoal-based decomposition (Eysenbach et al., 2019; Huang et al., 2019; Kim et al., 2021; Lee et al., 2022). At a high level, these planners construct a graph that encapsulates confident transitions between subgoals as edges and employ shortest-path algorithms to leverage the confident edges. However, there is a significant caveat: these graphs comprise only subgoals in the replay buffer, which the agent has achieved in prior. The oversight of not recording failed subgoals in history leads to wasteful expenditure of samples to repeatedly attempt similar unattained or even impossible subgoals rather than to explore novel ones, as exemplified in Figure 1.

In response to this challenge, we propose **B**readth-first **E**xploration on **A**daptive **G**rid (BEAG) for graph-based goal-conditioned RL. Our key idea is to manage both achieved and unattained subgoals on a grid graph, adaptively discretizing the goal space during the training process. Our subgoal management assesses the achievability of all the subgoals on the grid, including unattained ones, and then probes the subgoals in a planned order. This systemically prevents consecutive attempts to (currently) unachievable subgoals, whereas the previous method, e.g., (Lee et al., 2022), expend attempts to randomly ransack unexplored subgoals, as shown in Figure 1.

Specifically, we devise a breadth-first exploration strategy grounded on local adaptive grid refinement. This strategy first explores subgoals on a coarse grid and then refines the grid selectively around the local of unattained subgoals. The benefits are twofold: (i) it prioritizes broad probing on a coarse grid over extensive searching on a dense grid; and (ii) adaptively refines the grid around

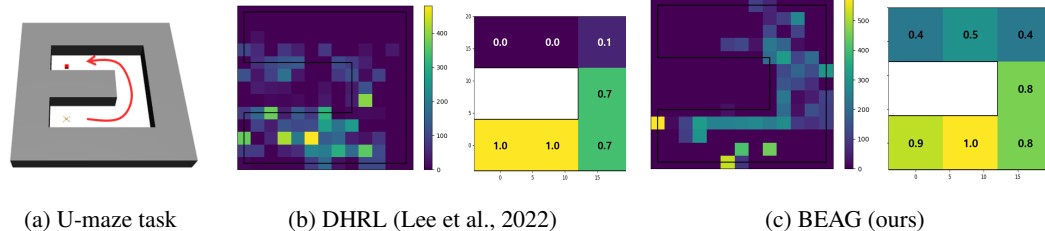

(a) U-maze task       (b) DHRL (Lee et al., 2022)       (c) BEAG (ours)

Figure 1: **Illustration of breadth-first exploration.** We compare the subgoal exploration strategies of DHRL (state-of-the-art) and BEAG (ours) in the 10-th training epoch for U-maze task, depicted in Figure 1a. In Figure 1b and 1c, each left plot summarizes the statistics on the attempted subgoals, and each right plot presents the agent's coverage in terms of the success rate of reaching goals in a specific region. DHRL expend a substantial number of attempts at impossible subgoals (on the wall), whereas BEAG spends virtually zero attempts on them. This demonstrates the efficiency of BEAG conducting the breadth-first exploration.

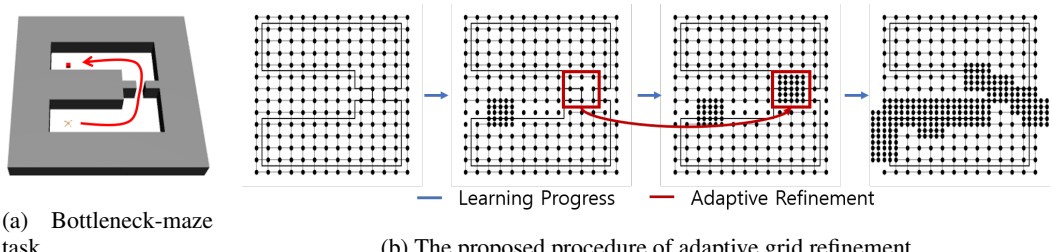

(a) Bottleneck-maze task

(b) The proposed procedure of adaptive grid refinement

Figure 2: **Local adaptive grid refinement.** In Bottleneck-maze task of Figure 2a, we visualize how BEAG adaptively refines the grid of subgoals over training epochs (0, 10, 11, 40), initialized at a grid too coarse to complete the task, compared to the width of the bottleneck point. As shown in Figure 2b, our approach adaptively refines around informative subgoals, previously unattainable and thus disconnected.

more demanding parts, as elucidated in Figure 2. Our extensive experiments demonstrate the efficacy of our method, in which BEAG identifies adaptive grids tailored to various environments requiring heterogeneous grid resolutions, both at the level of individual environments and within different parts of a single environment.

Our main contributions are summarized as follows:

- We show that the previous graph-based planners overlook unattained subgoals in their graphs, and thus often misguide the agent to wastefully attempt to impossible subgoals.
- To address this inefficiency, we introduce a grid-based RL, called BEAG, which efficiently identifies both achieved and unattained subgoals through the breadth-first exploration based on the local adaptive grid refinement over the goal space.
- Our method outperforms state-of-the-art methods in a set of complex and long tasks thanks to its adaptability to various environments with differing grid resolution requirements.

## 2 RELATED WORKS

**Graph-based RL and graph management**   Graph-based RL has emerged as a promising framework for solving complex tasks (Huang et al., 2019; Eysenbach et al., 2019; Kim et al., 2021; Lee et al., 2022), where deep RL agents plan a path of manageable subgoals from a challenging goal while leveraging graph algorithms such as shortest path algorithms on a subgoal graph. Despite their remarkable advancements, they just discard unattained subgoals and construct graphs of achieved ones, randomly sampled from the replay buffer, inheriting the practice originated by (Huang et al., 2019; Eysenbach et al., 2019). This restricted and random construction of graphs inevitably omits records of unattainable subgoals, which are beneficial to avoid them in the future. Consequently,

this omission can lead deep RL agents to fruitlessly pursue impossible subgoals, thereby diluting the benefits offered by graph algorithms. To address this issue, we introduce a grid-based method to embrace both attained and unattained subgoals on a grid, systematically exploring them. Specifically, we introduce the breadth-first exploration on a local adaptive grid refinement, where the refinement mechanism is analogous to the principle in (Berger & Oliger, 1984).

**Structured exploration in RL**   For efficient RL, exploration strategies have been structured to exploit the prior knowledge on the environment. This body of work encompasses both theoretical (Combes et al., 2017; Ok et al., 2018) and empirical (Pathak et al., 2017; Tam et al., 2022) studies. In hierarchical RL, the prior methods to expand the set of confident subgoals (Zhang et al., 2020; Kim et al., 2021; Lee et al., 2022) are structured under the assumption that more distant goals are inherently more challenging. They prioritize subgoals adjacent to previously identified confident ones. However, they overlook another valuable piece of prior knowledge that the achievabilities of similar goals are likely to be similar. Hence, they are prone to expend similar attempts around an impossible subgoal even after experiencing consecutive failures, rather than exploring more broadly for novel attempts. In contrast, our proposed breadth-first exploration strategy takes into account the prior knowledge of the similarity, thereby promoting a more efficient exploration. While similar approaches to broaden exploration ranges have been explored in various RL methods, e.g., (Pathak et al., 2017; Tam et al., 2022), our work is, to the best of our knowledge, the first one tailored for goal exploration. In addition, our adaptive refinement provides further improvements.

Finally, we underscore that a number of the previous works (Eysenbach et al., 2019; Huang et al., 2019; Zhang et al., 2021; Gieselmann & Pokorny, 2021; Kim et al., 2023) in fact, bypassed the subgoal exploration problems by assuming *uniform initial state distribution over the feasible space*. This can be direct access to the area of achievable subgoals. In our experiment, we fixed the initial states instead to evaluate the efficiency of subgoal exploration. In turn, the most recent work (Kim et al., 2023) showed a very low performance, while BEAG solved a set of complex tasks while quickly identifying achievable subgoals.

## 3    PRELIMINARIES

### 3.1    GOAL-CONDITIONED REINFORCEMENT LEARNING.

**Learning framework**   Building upon previous research (Schaul et al., 2015; Andrychowicz et al., 2017), we begin with a goal-conditioned Markov decision process represented as $(\mathcal{S}, \mathcal{G}, \mathcal{A}, p, r, \gamma, H)$, where $\mathcal{S}$ and $\mathcal{G}$ are state space and goal space respectively and there is a mapping function $\phi : \mathcal{S} \to \mathcal{G}$ between them. $\mathcal{A}$ denotes the action space, and environment works by the transition dynamics $p(s'|s, a)$. Reward function is defined as $r(s, a, s', g) = -\mathbf{1}_{|\phi(s')-g|>\delta}$, where $\delta$ is a threshold to determine whether if the goal has been achieved. $\gamma$ denotes the discount factor and $H$ is the horizon. Our framework contains a policy denoted as $\pi(a|s, g; \theta)$, which is parameterized by neural networks with parameters represented as $\theta$. And the objective of our framework is to optimize $\pi$ in order to maximize the expected future outcome $\Sigma_{t=0}^{\infty} \gamma^t r(s_t, a_t, s_{t+1}, g)$.

**Hindsight experience replay**   Given the assumption of sparse rewards and long-horizon tasks, there exists an efficiency challenge in learning. To address this, many previous methods have incorporated Hindsight Experience Replay (HER) (Andrychowicz et al., 2017) into their training. HER involves modifying the goal by covering the achieved goals in the future during training. By doing so, it allows for more sample-efficient learning by making it possible to learn from failed attempts as if they were successful, particularly in long-horizon tasks. We also apply Hindsight Experience Replay to our learning framework for enhanced sample efficiency.

### 3.2    GRAPH-BASED REINFORCEMENT LEARNING.

**Planning on the graph**   Introducing a graph into goal-conditioned RL for complex and long-horizon tasks has shown remarkable performance improvement. In previous works, it has been shown that it is possible to shorten long horizons by changing the goal to one of the goals that the agent has already visited. Search on the Replay Buffer (SoRB) (Eysenbach et al., 2019) constructs a graph based on states randomly sampled from a replay buffer. SoRB connects edges with weights

estimated from the $Q$-function. Since the reward function mentioned above assigns a reward of $-1$ until reaching the goal, it aids in estimating temporal distance:

$$Dist(s \to g) = \log_\gamma(1 + (1-\gamma)Q(s, \pi(s,g)|g)), \tag{1}$$

which is the expected time step from state $s$ to goal $g$. Then, the path is generated by the shortest path finding algorithm, such as Dijkstra's algorithm.

However, this estimation may lead to errors when predicting long distances. To address this issue, previous methods (Eysenbach et al., 2019; Lee et al., 2022) have introduced a threshold on edge length, connecting nodes only within a certain distance. However, when a goal is assigned to a location that has not yet been explored, in other words, when a goal that is far from the current graph is given, it is essential for the goal to be connected to the current graph to use the shortest path algorithm. In such cases, the problem arises where incorrect estimations can lead to the generation of incorrect paths. Thus, we propose graph planning method dealing with the above problem in Section 4.

## 4 METHOD

To fully utilize the benefit from the graph-planner in graph-based RL, it is necessary to compose the subgoal graph with confident edges covering all the (feasible) goal space. To construct such a subgoal graph, we propose a *grid*-based RL, called breadth-first exploration on adaptive grid (BEAG). Our key idea is to systemically manage the statistics of failures and successes of subgoals on an adaptive grid. In Section 4.1, we explain how to construct and manage a grid. We begin with a regular grid, uniformly covering the goal space. While training, we mark unattained subgoals to avoid repetitve failures. Subsequently, we adaptively densify grid to facilitate fine-grained decision making. We manage the graph by iteratively marking and densifying. Additionally, We introduce goal shifting techniques to enhance exploration. In Section 4.2, We elaborate on the rationale behind adopting a grid graph and elucidate why our method demonstrates robust performance.

### 4.1 BREADTH-FIRST EXPLORATION ON ADAPTIVE GRID (BEAG)

**Initial grid**    We begin with a grid graph $\mathcal{H} = (\mathcal{V}, \mathcal{E})$ aligned with the $k$-dimensional goal space. To simplify the description, we use a uniform grid interval $n$ for each dimension: $\mathcal{V} = \{l, l+n, l+2n, ..., l+mn\}^k$, with the constraint $l+mn \leq u < l+(m+1)n$, where $l$ and $u$ represent the lower and upper bounds of $\mathcal{G}$, respectively. Subsequently, $\mathcal{E}$ is formed by connecting $i, j \in \mathcal{V}$ where the Euclidean distance between $i$ and $j$ is $n$. We assume that the cost of moving between two subgoals at an equal distance is similar, and thus, we use the Euclidean distance as the weights of the edges. To generalize to diverse environments, if the aforementioned assumption does not hold, the distance measure introduced in (1) can be employed as the weights of the edges. However, it is noteworthy that the Q-function may exhibit instability in predictions for untried transitions, potentially hindering exploration.

**Unattained subgoal marking**    During the planning process, we run a shortest path algorithm on the grid graph. In particular, we assign edge weights to identify unreachable subgoals. We implement this by introducing two hyperparameters $\tau_t, \tau_n$ which serve as threshold for the failure condition and count. To be specific, we assign the edge weights $w_{i,j}$ of the edge $(i,j)$ as follows:

$$w_{i,j} = \begin{cases} \infty & \text{if } n_j^a > \tau_n \text{ and } n_j^s = 0 \\ \|i-j\|_2 & otherwise \end{cases}, \quad i, j \in \mathcal{V}, (i,j) \in \mathcal{E}. \tag{2}$$

Here, $n_j^a$ denotes the number of attempts to the subgoal $j$ as the next subgoal. Next, $n_j^s$ denotes the number of successes for the agent to actually reach the subgoal $j$ in the attempt. Since we are using a grid graph, we can assume that if the next subgoal is not an unattinable subgoal, it can be achieved within proper time steps. Thus, instead of trying until the end of the episode to determine failure, we consider it a failure if the next subgoal is not achieved within $\tau_t$ time steps. In that case, we also assign the edge weights leading to that subgoal to $\infty$ for the duration of the episode. It is important to emphasize that postponing exploration for the subgoal occurs only within the current episode and modifications are made even if the attempt count exceeds $\tau_n$.

**Adaptive grid refinement**   It is worth noting that the performance of grid-based planning crucially dependent on the grid interval $n$, which is a hyperparameter to control the level of details for searching a trajectory. Similar to other hyperparameters, choosing an inappropriate grid interval may downgrade the performance, e.g., an overly large grid interval may lead to sub-optimal trajectories with coarse-grained transitions. To address this issue, we propose *local adaptive grid refinement* which provides robustness to the choice of grid interval via adaptively amplifying the grid resolution near the selected subgoals. Given the assumption that exploration in the failed region is more valuable than the already successful region, we choose the postponed subgoal that have failed more than $\tau_n$ times, as mentioned earlier in the weight criteria. To underscore our breadth-first exploration, we select the subgoal from those with the largest intervals, specifically those distant from surrounding subgoals. The grid refinement is achieved by adding a $5^k$ grid graph for $k$-dimensional goal space that are half the size of the previous grid interval around the selected subgoals. For instance, refining $(m, m, ..., m)$ would involve combining the grid graph $\{m - n, m - \frac{n}{2}, m, m + \frac{n}{2}, m + n\}^k$ to the existing grid graph. We simply monitor the number of visited subgoals, and if the count does not increase for several episodes, we determine that refinement is necessary and proceeded.

**Goal shifting**   As evidenced by prior works (Ecoffet et al., 2019), it is often informative to continue the exploration of an agent even after achieving the target goal. To this end, after an agent achieves the target goal, we shift the goal to one of the unattempted subgoals in the grid graph. We regularize the "hardness" of the new goal via prioritization of shift to the closest unattempted subgoal. This technique can be considered a more efficient form of exploration compared to prior methods that suggest a new goal via random noise addition that is more likely to suggest shifting the goal towards already explored subgoals in the later training phase of the agent.

## 4.2   DISCUSSION ABOUT BEAG

In this section, we further clarify the advantages of using our grid-based planning algorithm. First, the trajectories planned by the grid graph contains even the "novel subgoals", which represent subgoals that have not been explored by the agent in prior. This provides a useful cue for the agent to explore unattempted trajectories instead of overly concentrating on failed attempts in the past. Previous methods struggled to generate diverse path when given goal located outside the graph. As shown in Figure 1b, they often ended up generating paths similar to ones that had failed before, causing repeated failures. In contrast, within our grid graph, we can generate many alternative routes that can lead to the goal. In addition, the above edge weight modification criteria encourages the exploration of new subgoals instead of repeating failures.

Furthermore, the subgoal trajectories generated from the grid-based planner is reasonable in the sense that the distance between the subgoals are regularized to be within a reachable distance. This incorporates our prior knowledge about the environment that the agent likely cannot reach a far-away subgoal provided by the planner. In contrast, existing graph-based planners often set the final goal as the subgoals for the agent, which is likely unreachable during the initial training phase.

Finally, the grid graph also explicitly keeps track of failed attempts via subgoals and edges that represent unattainable subgoals and transitions, respectively. This complements the inherent bias of training agents with hindsight experience replay that promotes the agent to waste attempts on unattainable subgoals. This happens because the goal in the learning data gets replaced with one of the states achieved during the trajectory, causing a delay in learning from failed goals. By addressing failed goals during the planning phase rather than the network training phase, we can effectively avoid repetitive failures and even ensure the efficiency of learning through HER.

## 5   EXPERIMENTS

### 5.1   EXPERIMENTAL SETUP

**Baselines**   We compare our method, BEAG, with the state-of-the-arts of graph-based RL algorithms in the followings:

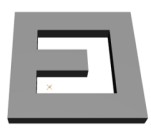 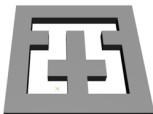 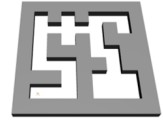 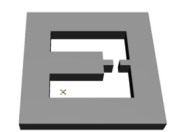 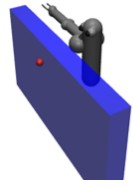

(a) U-maze   (b) Π-maze   (c) Complex-maze   (d) Bottleneck-maze   (e) Reacher3D

Figure 3: **AntMaze and Reacher3D environments.** We evaluate graph-based RL methods in the set of MuJoCo environments depicted above, in challenging setups with sparse rewards over long horizon. They include navigation tasks in various maps, and robot-arm manipulation task with obstacle to reach the target position, where the subgoal exploration is crucial as the agents have no access to any information on the maps except their maximum sizes.

- DHRL (Lee et al., 2022): DHRL constructs a graph from the replay buffer based on the FPS algorithm, which aids in generating an uniform graph. Then, it follows a shortest path from the current state to the subgoal generated by high-level policy.

- PIG (Kim et al., 2023): PIG also constructs a graph based on the FPS algorithm and employs imitation learning to ensure that the same action is taken for each subgoal along a shortest path. PIG employs random initialization during the training process, which is highly advantageous in the graph-based method due to its ease in collecting states for graph construction. For a fair comparison, we report results using fixed initialization during the training process.

- HIGL (Kim et al., 2021): HIGL constructs a graph from the replay buffer by considering both coverage and novelty. In addition, it employed a shortest-path algorithm to find a path and obtained a restricted subgoal using an adjacency network. While HIGL operates in a dense reward setting for their optimal performance, for a fair comparison, we report experimental results in a sparse reward setting.

- HIRO (Nachum et al., 2018): HIRO is a vanilla HRL method that has a 2-level learnable hierarchical policy and does not use graphs for training or inference.

**Training setups** Goal-conditioned RL has various training setups depending on the initial state space and goal space, each of which serves a unique purpose. For instance, there can be a fixed state space and a random goal space, where the objective is to be able to reach anywhere from the fixed location. If we use fixed goals for training in this scenario, the agent may quickly reach specific locations but this does not necessarily imply the ability to go anywhere. On the other hand, considering random initial states and goal spaces implies the objective of being able to go from anywhere to anywhere, regardless of the starting point. However, especially in graph-based RL, utilizing a random initial state allows for data collection across the entire environment without the need for exploration. This makes it a less challenging environment, and the scope of its utilization becomes limited. Therefore, we conducted the overall experiments in a learning environment similar to DHRL, featuring a fixed initial state and random goals. To be detailed, we implement our networks based on the TD3 algorithm (Fujimoto et al., 2018), and employ Dijkstra's algorithm to compute the shortest path in the graph. Additionally, it is noteworthy that we consistently set the grid interval to 2 for all maps in the AntMaze environment.

**Evaluation** Our primary evaluation metric for RL algorithms is the *success rate*, a widely adopted metric in prior works (Lee et al., 2022; Kim et al., 2021; 2023). Specifically, the success rate is the ratio of successfully achieving the most challenging goal in 10 trials. Given our method's emphasis on exploration, examining the success rate's improvement with fewer environment steps becomes crucial and forms a key point of interest.

**Environments** Our experiments were conducted on a set of challenging tasks with sparse rewards over long horizon, by configuring the MuJoCo environments (AntMaze and Reacher). Specifically, the following environments are considered, and visual representations are provided in Figure 3.

- {U, π, Complex, Bottleneck}-maze: The ants simulate reaching goals in a variety of mazes. Each map has a different size (U, Bottleneck : 24 × 24, π: 40 × 40, complex: 56 × 56)

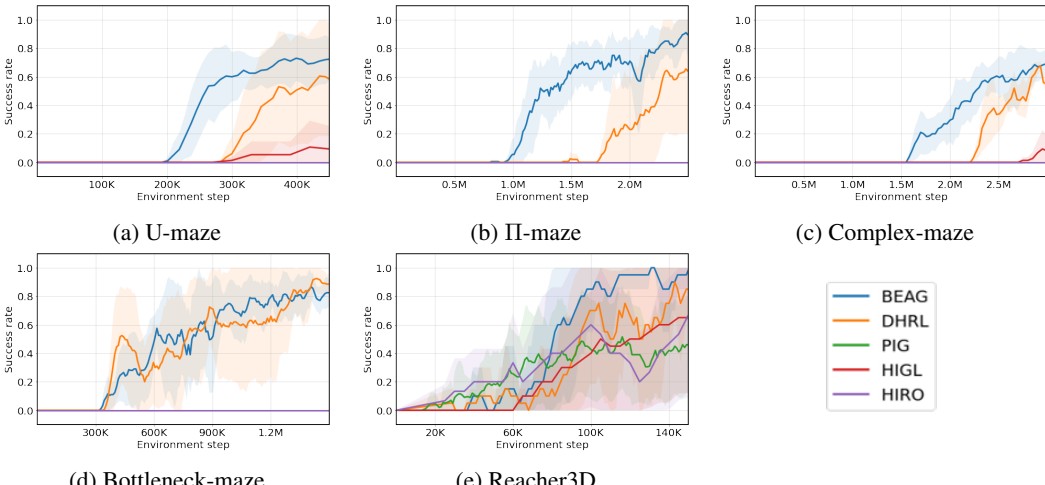

(a) U-maze  (b) Π-maze  (c) Complex-maze

(d) Bottleneck-maze  (e) Reacher3D

Figure 4: **Success rates in various environments (with random goal).** We report the average success rate as a line and the standard deviation as a shaded region for the 4 random seeds. We note that PIG (Kim et al., 2023) assumes a fixed initial state, and HIGL (Kim et al., 2021) operates under a sparse reward setting for a fair comparison.

and structure, possibly conjectured by its name. For instance, Bottleneck-maze, originated in (Lee et al., 2022), has an intriguing structure, where the maze is divided into two parts by a narrow gate or bottleneck between them, and thus the agent needs to recognize the gate in order to navigate the ant everywhere.

- Reacher3D (with a wall): The robotic arm simulates moving the tip of the hand to the target position, which can be blocked by a wall, as shown in Figure 3e.

## 5.2 COMPARATIVE RESULTS

As evident in Figure 4a, 4b, and 4c, our approach BEAG consistently outperforms the state-of-the-art baselines. The advantage of BEAG is more pronounced in terms of success rate on a larger map, i.e., the breadth-first exploration is more advantageous in larger maps ($\pi$-maze and Complex-maze). The reason that baselines require more samples to achieve the challenging goals is that it omits records on impossible subgoals in the graph while training the neural high-level agent to learn the impossible subgoals implicitly. In contrast, BEAG's breadth-first search systemically identifies the impossible subgoals on the grid. Additionally, it is important to highlight the relatively high variance observed in DHRL. This variability can be attributed to the utilization of randomly assigned goals during training, which can be significantly influenced by the initial random seed. As elucidated in Figure 1, BEAG employs breadth-first exploration, allowing it to construct a map understanding independent of the intricacy of randomly assigned goals.

## 5.3 FIXED GOAL ENVIRONMENT

We have conducted experiment to demonstrate a clear advantage of BEAG. Specifically we compare BEAG and DHRL in a U-maze environment, where the initial state and goal are fixed at the bottom-left and top-left corners, respectively. This environment is dedicated to evaluating the efficiency of exploration. As illustrated in Figure 5a, BEAG reached the goal much faster than DHRL, thanks to the breadth-first exploration. The relatively small gap between BEAG and DHRL in the random goal setting may suggest that DHRL heavily relies on random goal generation for subgoal exploration. In this experiment, we also report *coverage*, which is the averaged success rate of uniformly sampled goals, in which we sample 10 goals per unit-size cell over the entire goal space to make the sampling more uniform. As depicted in Figure 5b, DHRL exhibits a lack of coverage increase under the fixed goal setting, indicating a complete absence of exploration. In Figure 5c and 5d, we report the success rates for specific region during training, our approach stands out in its ability to explore diverse paths leading to challenging goals, owing to its management of unattained subgoals, while DHRL encounters difficulties in exploring novel areas. Furthermore, we observe that BEAG outperforms

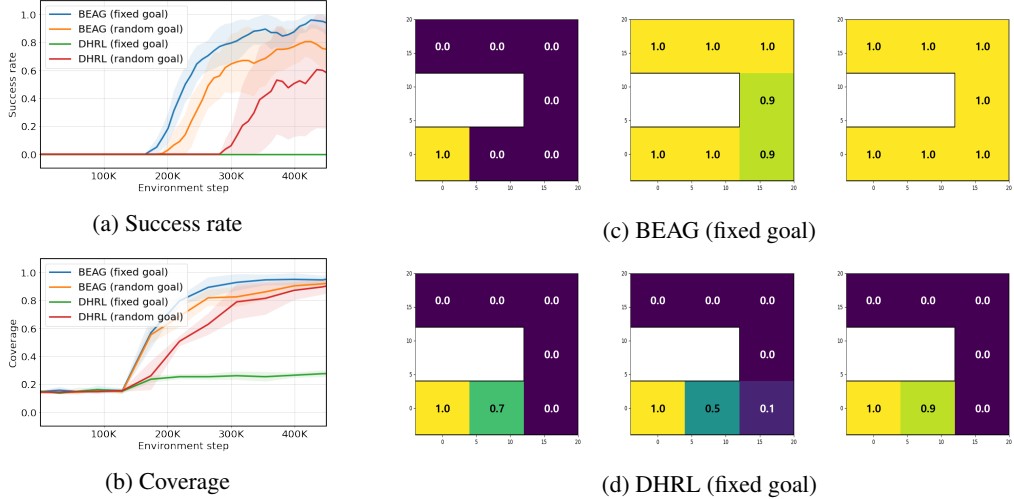

(a) Success rate

(c) BEAG (fixed goal)

(b) Coverage

(d) DHRL (fixed goal)

Figure 5: **U-maze with a fixed goal.** We measure the (a) average success rate and (b) average coverage for each region in U-maze, where the initial state and goal are fixed at the bottom-left and top-left corners, respectively, during the training phase. We also visualize the coverage of each region at 0, 20, and 40 epochs (120K, 300K, and 480K environment steps), respectively (c, d).

in the fixed goal setting compared to the random goal setting, as the continual input of the same goal during training facilitates the effective removal of unattained subgoals along the path toward the goal.

## 5.4 BOTTLENECK-MAZE ENVIRONMENT

We design an experiment to demonstrate the ability of grid refinement to address the challenge of a bottleneck environment in Figure 4d, which can be difficult to navigate on the grid graph. As shown in Figure 2, with the initialized grid graph, our algorithm can only fail for the upper half of the map. BEAG rapidly completes the exploration of the lower part through breadth-first exploration and refines the outer regions of the map for additional exploration. After refinement, it can generate paths even for areas where it previously failed, demonstrating the scalability and robustness to the hyperparameter of our algorithm. Despite disadvantage of narrow bottleneck for grid, BEAG demonstrates performance comparable to DHRL in comparison.

## 5.5 ABLATION STUDY

**Goal Shifting** Goal shifting is a technique that helps the agent engage in additional exploration by replacing an achieved goal with a nearby explorable goal. This reduces wasted episodes where easy goals are repeatedly provided in settings with randomly given goals. As a result, the average success rate has improved, but it is important to pay closer attention to the variance. As shown in Figure 6 a significant reduction in variance is observed, emphasizing the learning stability of BEAG concerning random seeds, as mentioned in Section 5.2.

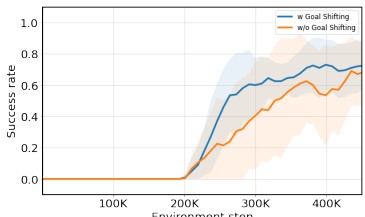

Figure 6: **Ablation study about Goal Shifting in U-maze.**

**Hyperparameter Choice** To identify failure, we utilize a count-based method with two thresholds: failure condition $\tau_t$ and failure count $\tau_n$. Figures 7a and 7b demonstrate that increasing the thresholds for both criteria requires more environment steps to achieve goals. Notably, a low success rate is observed at $\tau_n = 1$ in Figure 7b, attributed to a significant number of false negatives (attainable but marked), as illustrated in Figure 7e, especially when compared to Figure 7c. In Figure 7a, it can be observed that although the performance is favorable at $\tau_t = 50$, however, false negatives still occur under this condition as demonstrated in Figure 7d. To ensure stability, we employed the thresholds ($\tau_t = 100, \tau_n = 3$) in all other experiments.

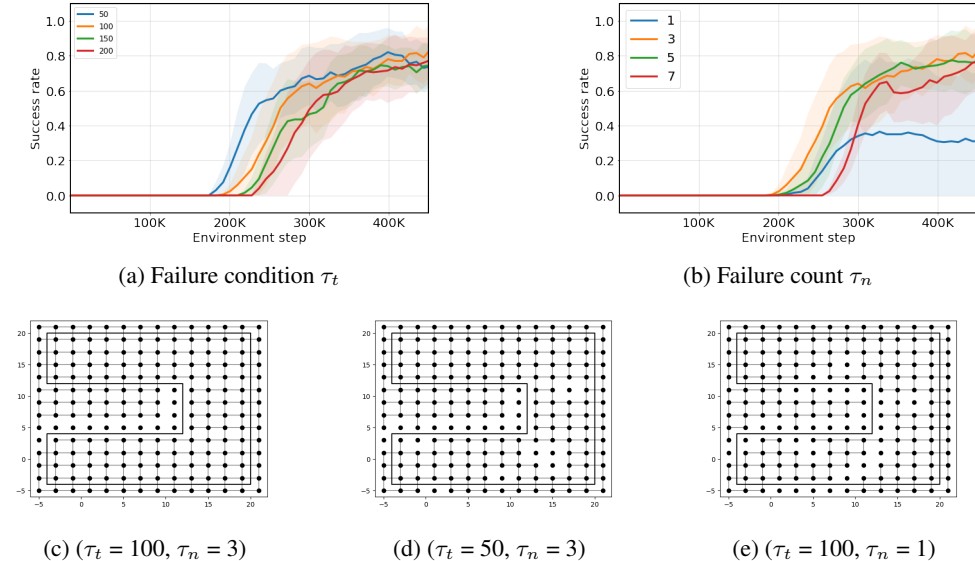

(a) Failure condition $\tau_t$        (b) Failure count $\tau_n$

(c) ($\tau_t = 100$, $\tau_n = 3$)     (d) ($\tau_t = 50$, $\tau_n = 3$)     (e) ($\tau_t = 100$, $\tau_n = 1$)

Figure 7: **Hyperparameter choice in U-maze.** We measure the average success rate of BEAG on different (a) failure condition $\tau_t$ and (b) failure count $\tau_n$. We also visualize the grid graphs at the last timestep (c, d, e). We visualize the grid graph by removing edges directed to the marked unattained subgoals.

**Design Choice** To construct the graph, the grid graph is employed due to its uniform distribution, ensuring the presence of a nearby subgoal even with a limited number of nodes, irrespective of the given goal. We present experimental results applying our proposed method to a graph composed of uniformly randomly sampled nodes across the entire map. As illustrated in Figure 8, outcomes from random nodes resulted in failure with a comparable number of nodes (200). Furthermore, with an increase in the number of nodes (400, 600), a need to postpone nodes also escalates, demanding more environment steps for success.

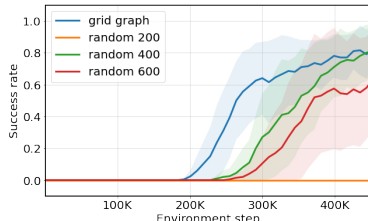

Figure 8: **Empirical effectiveness of design choice in U-maze.**

## 6 CONCLUSION

In this work, we introduce BEAG, a novel approach that places a strong emphasis on sample-efficient exploration to enhance goal achievement. Our experimental findings showcase the rapid expansion of exploration, facilitated by predictive techniques applied to nodes previously considered unreachable within a grid graph framework. BEAG not only surpasses the performance of prior state-of-the-art methods but also introduces a fresh perspective to the realm of graph-based reinforcement learning. While we have applied a straightforward grid refinement approach to a grid graph in this study, we anticipate the existence of more sophisticated heuristics for adaptive grid graph construction and the selection of candidate refinements in the future.

**Limitation** BEAG has demonstrated significant performance improvements in the context of sparse and long-horizon tasks, specifically illustrated through the simplified example of AntMaze navigation with a 2-dimensional goal space. However, it is expected to face limitations when applied to high-dimensional datasets such as images or language due to the inherent complexities involved. We believe that further research into representation learning for handling high-dimensional data can address these challenges. Additionally, while our approach primarily utilized grids, incorporating representations would necessitate starting from graph construction. We leave these aspects as future works.

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

# A    ALGORITHM

---

**Algorithm 1** Overview of BEAG

---

**Input:** total training episode $n_{\text{epi}}$, initial random step $n_{\text{rand}}$, refinement frequency $n_{\text{ref}}$, total training step $t_{\text{step}}$, failure count threshold $\tau_n$, failure condition threshold $\tau_t$, environment Env, policy $\pi$, state-goal mapping function $\phi$

**for** $ep \leftarrow 1$ **to** $n_{\text{epi}}$ **do**
    Env.reset()
    **if** $ep = n_{\text{rand}}$ **then**
        Construct a grid graph $\mathcal{H}(\mathcal{V}, \mathcal{E})$
        Marked unattained subgoal set $\mathcal{V}_u \leftarrow \emptyset$
    **else if** $ep > n_{\text{rand}}$ **then**
        **if** $ep \% n_{\text{ref}} = 0$ **then**
            $\mathcal{H}, \mathcal{V}_u \leftarrow$ Grid Refinement($\mathcal{H}, \mathcal{V}_u$)
        **end if**
        $\mathcal{H}'(\mathcal{V}', \mathcal{E}') \leftarrow \mathcal{H}$
    **end if**
    **for** $t \leftarrow 1$ **to** $t_{\text{step}}$ **do**
        **if** $ep < n_{\text{rand}}$ **then**                                ▷ Initial random stage
            $a_t \leftarrow$ random.unifrom(Env.action)
        **else**
            **if** episode step = 0 **then**
                $\mathcal{P} \leftarrow$ Dijkstra's Algorithm($\mathcal{H}', \phi(s_0), g$): $(\phi(s_0), wp_1, wp_2, ..., g)$
                waypoint index $p \leftarrow 1$; tracking time $t_{tr} \leftarrow 0$
            **end if**
            **if** achieved goal $g$ **then**
                $g \leftarrow$ Goal Shifting($\mathcal{H}', g$)
                $\mathcal{P} \leftarrow$ Dijkstra's Algorithm($\mathcal{H}', \phi(s_t), g$): $(\phi(s_t), wp_1, wp_2..., g)$
                $p \leftarrow 1$; $t_{tr} \leftarrow 0$
            **else if** achieved waypoint $wp_p$ **then**
                $n^a_{wp_p} \leftarrow n^a_{wp_p} + 1$
                $n^s_{wp_p} \leftarrow n^s_{wp_p} + 1$
                $p \leftarrow p + 1$; $t_{tr} \leftarrow 0$
            **else if** $t_{tr} > \tau_t$ **then**
                $n^a_{wp_p} \leftarrow n^a_{wp_p} + 1$
                $\mathcal{H}' \leftarrow$ Remove Subgoal($\mathcal{H}', wp_p$)
                **if** $n^a_{wp_p} > \tau_n$ and $n^s_{wp_p} = 0$ **then**
                    $\mathcal{H} \leftarrow$ Remove Subgoal($\mathcal{H}, wp_p$)
                    $\mathcal{V}_u \leftarrow \mathcal{V}_u \cup \{wp_p\}$
                **end if**
                $\mathcal{P} \leftarrow$ Dijkstra's Algorithm($\mathcal{H}', \phi(s_t), g$): $(\phi(s_t), wp_1, wp_2..., g)$
                $p \leftarrow 1$; $t_{tr} \leftarrow 0$
            **end if**
            $a_t \leftarrow \pi(s_t, wp_p)$
        **end if**
        Env.step($a_t$)
        Train policy $\pi$
        $t_{tr} \leftarrow t_{tr} + 1$
    **end for**
**end for**

---

---

**Algorithm 2** Training policy

---

**Input:** replay buffer $\mathcal{B}$, maximum her step $h_{max}$

sample $D \leftarrow (s_t, sg_t, a_t, r(s_{t+1}, sg_t), s_{t+1}) \in \mathcal{B}$
relabel $sg_t \leftarrow s\hat{g}_t = \phi(s_{t+t_h})$ where $t_h \sim \text{Uniform}([0, \min(t_{total} - t, h_{max})])$
update $\pi$ using $D$

---

**Algorithm 3** Grid Graph Construction

---

**Input:** grid interval $n$, goal space dimension $m$, lower bound of goal space $l$, upper bound of goal space $u$
**Output:** graph $\mathcal{H}(\mathcal{V}, \mathcal{E})$

vertex set $\mathcal{V} \leftarrow \emptyset$; edge set $\mathcal{E} \leftarrow \emptyset$
**for** $m \leftarrow 1$ **to** $k$ **do**
    $\mathcal{V}_m \leftarrow \emptyset$
    $x \leftarrow l$
    **while** $x < u$ **do**
        $\mathcal{V}_m \leftarrow \mathcal{V}_m \cup \{x\}$
        $x \leftarrow x + n$
    **end while**
**end for**
$\mathcal{V} \leftarrow \mathcal{V}_1 \times \mathcal{V}_2 \times ... \times \mathcal{V}_k$
**for** $i \in \mathcal{V}$ **do**
    $h_i \leftarrow n$                                           ▷ grid interval
    $n_i^a \leftarrow 0$                                      ▷ attempt count
    $n_i^s \leftarrow 0$                                      ▷ success count
**end for**
**for** $i, j \in \mathcal{V}$ **do**
    **if** $\|v_i, v_j\|_2 \leq n$ **then**
        $w_{i,j} \leftarrow n; w_{j,i} \leftarrow n$
        $\mathcal{E} \leftarrow \mathcal{E} \cup \{(i,j), (j,i)\}$
    **end if**
**end for**
**return** $\mathcal{H}(\mathcal{V}, \mathcal{E})$

---

**Algorithm 4** Grid Refinement

---

**Input:** graph $\mathcal{H}$, marked unattained subgoal set $\mathcal{V}_u$
**Output:** expanded graph $\mathcal{H}$, marked unattained subgoal set $\mathcal{V}_u$

**if** the number of succeeded subgoals in $\mathcal{V}$ is increased **then**
    **return** $\mathcal{H}, \mathcal{V}_u$
**end if**
**if** $\mathcal{V}_u$ is empty **then**
    **return** $\mathcal{H}, \mathcal{V}_u$
**end if**
$d \leftarrow \max_{i \in \mathcal{V}_u} h_i$
$i \leftarrow$ uniformly random sample from $\mathcal{V}_u$ where $h_i == d$
$\mathcal{V}_u \leftarrow \mathcal{V}_u \setminus \{i\}$
$h_i \leftarrow h_i / 2$
**for** $j \in \mathcal{G}_{i,h_i}^{5 \times 5}$ **do**
    **if** $j \notin \mathcal{G}$ **then**
        $\mathcal{V} \leftarrow \mathcal{V} \cup j$
    **end if**
    Add edges between $j$ and $v \in \mathcal{V}$ which $\|j, v\|_2 \leq h_i$
**end for**
**return** $\mathcal{H}, \mathcal{V}_u$

---

---

**Algorithm 5** Goal Shifting

---

**Input:** graph $\mathcal{H}$, goal $g$
**Output:** goal $g$

$\mathcal{V}_s \leftarrow \{i | i \in \mathcal{V}, \ n_i^a = 0\}$
**if** $\mathcal{V}_s \neq \emptyset$ **then**
    $g \leftarrow \underset{i}{\mathrm{argmin}} \, \mathrm{dist}(\text{Dijkstra's algorithm}(g, i)) \text{ for } i \in \mathcal{V}_s$
**end if**
**return** $g$

---

---

**Algorithm 6** Remove Subgoal

---

**Input:** graph $\mathcal{H}(\mathcal{V}, \mathcal{E})$, subgoal $j$
**Output:** modified graph $\mathcal{H}(\mathcal{V}, \mathcal{E})$

**for** $(i, j) \in \mathcal{E}$ **do**
    $w_{i,j} \leftarrow \infty$
**end for**
**return** $\mathcal{H}$

---

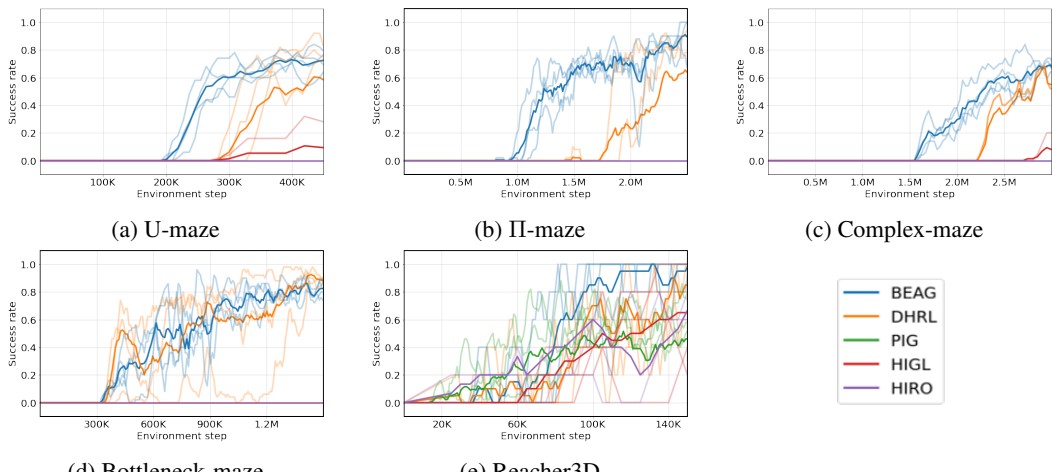

(a) U-maze        (b) Π-maze        (c) Complex-maze

(d) Bottleneck-maze        (e) Reacher3D

Figure 9: **Success rates across individual seeds.** We depicted individual seed values with lighter lines and emphasized the mean with bolder lines.

## B  SUCCESS RATES ACROSS INDIVIDUAL RANDOM SEEDS

We present a refined representation of experiment results in Figure 4. In contrast to the original graph depicting the mean and standard deviation, this visualization subtly overlays individual seeds for enhanced clarity.

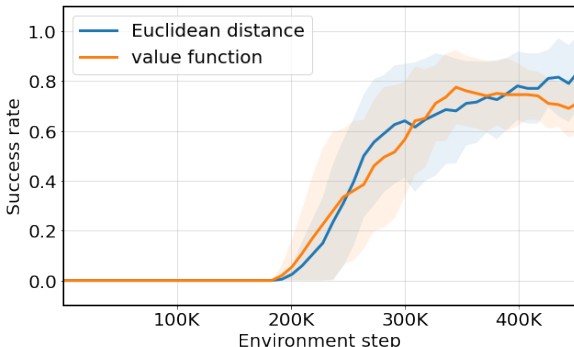

Figure 10: **Effectiveness of the distance choice in U-maze.** *value function* represents the results of BEAG conducted using a graph where the weight of edges are assigned based on the distance in (1).

## C DESIGN CHOICE FOR THE WEIGHT OF EDGES

BEAG operates under the assumption that the cost of moving between two subgoals at an equal distance is similar, determining the weight of edges based on Euclidean distance. However, this assumption is not universally valid. In more general environments, BEAG can utilize the distance defined in (1) for the weight of edges. Figure 10 demonstrates BEAG's comparable performance with this distance. Nevertheless, while subgoals in previous approaches consist of previously visited subgoals, ensuring accurate predictions, our graph includes unseen subgoals. As a result, in the initial stages, (1) may lead to unexpected predictions for unseen edges, potentially generating unfavorable paths to the goal.

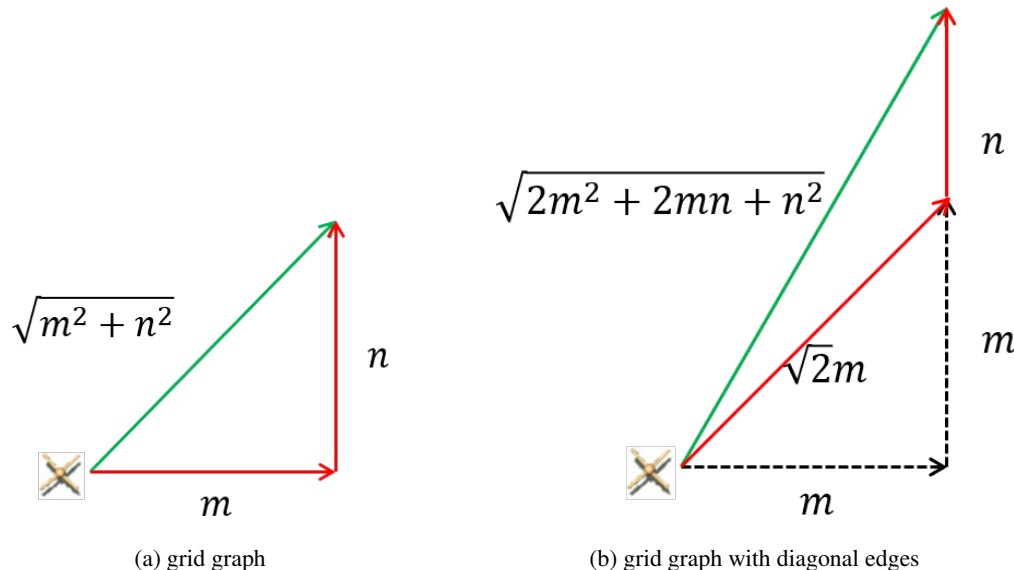

(a) grid graph                    (b) grid graph with diagonal edges

Figure 11: **Suboptimality of grid graph.** We calculate the suboptimality of the path on the 2D environment for (a) the proposed grid graph and (b) the proposed grid graph with diagonal edges. The red path represents the path generated by the grid graph, while the green path corresponds to the straight path.

## D ANALYZING SUBOPTIMALITY OF THE GRID PATH

We are analyzing the issue of suboptimality in our grid path planning. BEAG involves planning paths that limit movement to a single axis, resulting in the creation of suboptimal paths. For instance, within the 2D AntMaze environment, paths are confined to movement along either the x or y-axis. This constraint causes delays in reaching diagonal goals compared to the straight path, contributing to longer travel times. Our analysis specifically focuses on quantifying the suboptimality of these grid paths, especially in worst-case scenarios. We assume a 2D obstacle-free environment and that a perfectly trained low-level policy is available, asserting that the time required for movement is directly proportional to the Euclidean distance.

As shown in Figure 11a, the time for the path generated by the grid path is $m + n$, while the straight path takes $\sqrt{m^2 + n^2}$. The suboptimality $\frac{\sqrt{m^2+n^2}}{m+n}$ reaches its minimum value at $n = m$, approximately $\frac{\sqrt{2}}{2} \approx 0.707$. We also analyzing suboptimality when introducing diagonal edges into the grid graph, as depicted in Figure 11b. In this case, paths generated by the grid graph with diagonal edges take a time of $\sqrt{2}m + n$, while the straight path takes a time of $\sqrt{2m^2 + 2mn + n^2}$. The suboptimality $\frac{\sqrt{2m^2+2mn+n^2}}{\sqrt{2}m+n}$ is calculated as $\frac{\sqrt{2+\sqrt{2}}}{2} \approx 0.924$ when $n = \sqrt{2}m$. Our analysis suggests that simply adding diagonal edges when constructing the grid graph can significantly reduce the suboptimality of paths. However, it is important to note that diagonal edges have a distance $\sqrt{2}$ times that of the existing grid edges. This introduces a potential trade-off, as additional steps may be necessary for training the low-level policy.

# E HYPERPARAMETER CHOICE

When evaluating the previous Graph-based RL method, we used the same hyperparameters as used in their papers. And, we conducted additional tuning the number of landmarks for a fair comparison.

Table 1: **Hyperparameter for DHRL and BEAG.**

|  | DHRL | Ours |
|---|---|---|
| initial episodes without graph planning | 75 | - |
| gradual penalty | 1.5-5.0 | - |
| high-level train freq | 10 | - |
| Frontier-based Goal Shifting | $\{\pi, \text{Complex}\}$-maze | - |
| number of landmarks | 300-600 | 196-900 |
| hidden layer | (256, 256) | (256, 256) |
| actor lr | 0.0001 | 0.0001 |
| critic lr | 0.001 | 0.001 |
| $\tau$ | 0.005 | 0.005 |
| $\gamma$ | 0.99 | 0.99 |
| batch size | 1024 | 1024 |
| target update freq | 10 | 10 |
| actor update freq | 2 | 2 |

Table 2: **Hyperparameter for PIG.**

|  | Reacher | U-maze | $\pi$-maze | Complex-maze |
|---|---|---|---|---|
| Initial random trajectories | 20k | 100k | 400k | 800k |
| Number of nodes in a graph | 80 | 400 | 500 | 500 |
| Balancing coefficient $\lambda$ | 0.0001 | 0.001 | 0.001 | 0.001 |
| Skipping temperature $\alpha$ | 10.0 | 10.0 | 10.0 | 10.0 |
| Hindsight relabelling range | 50 | 200 | 200 | 200 |
| Action L2 | 0.01 | 0.5 | 0.5 | 0.5 |
| Action noise | 0.1 | 0.2 | 0.2 | 0.2 |
| clipping threshold for distances | 4.0 | 38.0 | 38.0 | 38.0 |

Table 3: **Hyperparameter for HIGL and HIRO.**

|  | HIGL | HIRO |
|---|---|---|
| higl-level $\tau$ | 0.005 | 0.005 |
| $\pi^{\text{hi}}$ lr | 0.0001 | 0.0001 |
| $Q^{\text{hi}}$ lr | 0.001 | 0.001 |
| high-level $\gamma$ | 0.99 | 0.99 |
| low-level $\tau$ | 0.005 | 0.005 |
| $\pi^{\text{lo}}$ lr | 0.0001 | 0.0001 |
| $Q^{\text{lo}}$ lr | 0.001 | 0.001 |
| low-level $\gamma$ | 0.95 | 0.95 |
| hidden layer | (128, 128) | (128, 128) |
| number of coverage landmarks $\gamma$ | 20-100 | - |
| number of novelty landmarks $\gamma$ | 20-400 | - |
| batch size | 128 | 128 |

