# OpenReview forum: "Breadth First Exploration in Grid-based Reinforcement Learning"
_ICLR.cc/2024/Conference — Submitted to ICLR 2024_

### Official Review · Reviewer_SMnh · 2023-10-25

**Soundness:** 3 good
**Presentation:** 2 fair
**Contribution:** 3 good
**Rating:** 5
**Confidence:** 4

**Summary:**

The paper introduces an algorithm for hierarchical planning. To plan the solution, the goal space is divided into a grid, and the shortest path is computed. During training, the reachability statistics for the visited edges are collected, and the graph is updated to reflect which transitions are achievable and which are not. This way, both exploration and the final quality of solutions benefit.

**Strengths:**

As far as I understand, the main idea is simple yet effective, especially in environments with complex dynamics but simple goal space, such as various ant-mazes. Because the planning is done by searching on a graph, it can deal even with an arbitrarily long horizon.

**Weaknesses:**

The description of the algorithm itself needs to be stated more clearly. How exactly do you collect the training data? Do you always choose the shortest path (according to the current state of the graph)? When do you switch subgoals? Do you have to replan at some point? It could be handled by either adding a small section that brings all the pieces in one place and describes the algorithm step-by-step or by adding a pseudocode. Without that, it is hard to understand your approach.

Since your edges have different lengths, I suppose that you use Dijkstra to find the shortest path, instead of BFS as you claim.

I agree that you show a clear advantage over DHRL, although the final difference is not very large. I suggest evaluating the algorithm on even harder instances to further highlight your superiority. Perhaps instances with a lot of blockers would do since you claim dealing with it to be one of your key strengths. How far can you push it, even in a toy environment?

The _Ablation study_ section seems a little ad-hoc. I suggest extending it or merging it with other experiments.

**Questions:**

How exactly do you collect the training data? Do you always choose the shortest path (according to the current state of the graph)? When do you switch subgoals? Do you have to replan at some point?

Do you ever remove edges from the graph, or is it simply enough to assign them $\infty$ weight?

How do you handle the initial stage of the training, when the agent has to learn the basics, e.g. how to move? In particular, I'm afraid that initially the agent will be unable to reach _any_ subgoal (barely move), which should result in labelling many goals close to the initial state as _unreachable_. This is not entirely destructive, since even if all the subgoals adjacent to the starting positions have a weight of $\infty$, the path will go through one of them anyway. However, this initial struggle may considerably bias the graph structure, which may affect further shortest paths (that will try to avoid the falsely failed areas). Is that indeed an issue and how do you handle it?

Also, apart from the initial bias, if any edge is marked with $\infty$ (possibly due to underfitting in a yet underexplored area), is there _any_ way of including it back to the graph?

Is it of any importance that you use a grid? I suppose it can be an arbitrary structure, like random samples. Does using the grid has advantages? It seems a little rigid.

During planning, you always select the shortest path to the goal in the graph. Does it mean that the agent is committed to this rigid grid-like movement, even if it could move diagonally? Do you see a way to adjust the structure of the graph to eventually allow for near-optimal trajectories? At least, trajectories without considerable inefficiencies?

It seems to me that this approach is useful if the goal space is low-dimensional. For instance, in 2 dimensions, like ant-maze, it is pretty efficient. However, if we consider any more complex spaces, e.g. ant-maze with the goal space equal to the observation space (which is quite natural in many scenarios), will it still work?

If the goal space is reduced, then the reachability of adjacent subgoals depends on the state in which the agent reaches each point, e.g. the exact position of its limbs. In particular, it may depend on the path that it took. How do you handle that?

In general, **I really like the high-level idea**, but in its current form, it is very hard to understand the essential details. Thus, I'm willing to increase my rating if my concerns are addressed.

---

> ### Author Response · Authors · 2023-11-19
>
> Dear Reviewer SMnh
>
> We would like to express our sincere gratitude for your valuable comments and insightful feedback. Your comment has greatly enriched our work and provided us with valuable perspectives. We address your comments one by one in what follows.
> ___
>
> **Q1 [Weakness1 & Question1]  The description of the algorithm itself needs to be stated more clearly. How exactly do you collect the training data? Do you always choose the shortest path (according to the current state of the graph)? When do you switch subgoals? Do you have to replan at some point? It could be handled by either adding a small section that brings all the pieces in one place and describes the algorithm step-by-step or by adding a pseudocode. Without that, it is hard to understand your approach.**
>
> **A1** We appreciate your meticulous reviews on the description of the proposed method, BEAG. Based on your questions and comments, we have substantial revised the draft (particularly, Appendix A) and added a comprehensive pseudocode of our algorithm, as you suggested. We individually respond to each question for clarification in the following:
>
> &nbsp; ***Q1-1. How exactly do you collect the training data?***
>
> &nbsp; ***A1-1.*** Initially, we collect the training data from the random rollout phase where the agent proceeds with actions entirely at random. After the random rollout phase, we collect the training data using the policy network with BEAG.
>
> &nbsp; ***Q1-2. Do you always choose the shortest path (according to the current state of the graph)?***
>
> &nbsp; ***A1-2.*** Yes, we always use the shortest path.
>
> &nbsp; ***Q1-3. When do you switch subgoals?***
>
> &nbsp; ***A1-3.*** We switch subgoals when we reach the subgoal. More precisely, when a subgoal falls within $\delta$. (i.e., $\lVert \phi(s’)-g \rVert < \delta$ )
>
> &nbsp; ***Q1-4. Do you have to replan at some point?***
>
> &nbsp; ***A1-4.*** We re-plan the path when we fail to achieve the subgoal in the failure condition threshold (100) step.
> ___
> **Q2. [Weakness2] Since your edges have different lengths, I suppose that you use Dijkstra to find the shortest path, instead of BFS as you claim.**
>
> **A2.**
> It appears there might be a misunderstanding. What we mean by breadth-first “exploration” is not aligned with the breadth-first search algorithm to compute the shortest path. It means that BEAG prefers broadly exploring bypasses for the currently unattainable subgoals than meticulously ransacking the local of them. We will further clarify the meaning of the breadth-first exploration in the revision. We thank the reviewer for pointing out the potential confusion of our expression.
> ___
> **Q3. [Weakness3] I agree that you show a clear advantage over DHRL, although the final difference is not very large. I suggest evaluating the algorithm on even harder instances to further highlight your superiority. Perhaps instances with a lot of blockers would do since you claim dealing with it to be one of your key strengths. How far can you push it, even in a toy environment?**
>
> **A3.**
> As requested, we have added an additional experiment and discussion, showcasing a clear advantage of BEAG over DHRL (See Figure 5 in Section 5.3). Specifically, we compare BEAG and DHRL in a U-maze environment, where the initial state and goal are fixed at the bottom-left and top-left corners, respectively. This environment is dedicated to evaluating the efficiency of exploration in various algorithms. BEAG reached the goal much faster than DHRL, thanks to the breadth-first exploration. The relatively small gap between BEAG and DHRL in the random goal setting may suggest that DHRL (and others) heavily relies on random goal generation for subgoal exploration. We hope the additional experiment and discussion address your question.
> ___
> **Q4. [Weakness4] The Ablation study section seems a little ad-hoc. I suggest extending it or merging it with other experiments.**
>
> **A4.**
> We appreciate your valuable suggestions on additional experiments. As suggested, in Figure 7, we conducted ablation studies on BEAG with different values of ($\tau_t$: [50, 100, 150, 200]) and ($\tau_n$: [1, 3, 5]).  As expected, both $\tau_t$ and $\tau_n$ (determining the thresholds to avoid subgoals) need to be set at minimal values to reduce attempts at impossible subgoals. However, exceedingly small values for $\tau_t$ and $\tau_n$ may accidentally identify attainable subgoals as impossible ones and possibly degenerate performances in early phases, as depicted in Figure 7 (d) and (e). This observation provides a useful guideline for the hyperparameter choice of our method. We hope that these additional experiments address the reviewer's concern.
> ___
> **Q5. [Question 2] Do you ever remove edges from the graph, or is it simply enough to assign them $\infty$ weight?**
>
> **A5.**
> For ease of implementation and presentation, we simply indicate the removed edge with infinite weight.

---

> > ### Author Response · Authors · 2023-11-19
> >
> > **Q6. [Question 3] How do you handle the initial stage of the training, when the agent has to learn the basics, e.g. how to move? In particular, I'm afraid that initially, the agent will be unable to reach any subgoal (barely move), which should result in labeling many goals close to the initial state as unreachable. This is not entirely destructive, since even if all the subgoals adjacent to the starting positions have a weight of \inf, the path will go through one of them anyway. However, this initial struggle may considerably bias the graph structure, which may affect further shortest paths (that will try to avoid the falsely failed areas). Is that indeed an issue and how do you handle it?**
> >
> > **A6.**
> > We appreciate the opportunity to clarify our algorithm based on your concern. BEAG begins with a roll-out phase where the agent proceeds with actions entirely at random without training, as DHRL does. This is to obtain a certain size of replay buffer and to let the agent learn basic movement. This mitigates the potential issue you mentioned.
> > ___
> >
> > **Q7. [Question 4] Also, apart from the initial bias, if any edge is marked with \inf (possibly due to underfitting in a yet underexplored area), is there any way of including it back to the graph?**
> >
> > **A7.**
> > Yes, there is. By the grid refinement procedure (illustrated in Figure 2), a subgoal which has been removed accidentally or correctly is randomly selected and restored into the graph.
> > ___
> >
> > **Q8. [Question 5] Is it of any importance that you use a grid? I suppose it can be an arbitrary structure, like random samples. Does using the grid has advantages? It seems a little rigid.**
> >
> > **A8.**
> > Thank you for your question on the rationale for choosing the grid instead of random graphs.
> > An advantage of the proposed method is to avoid excessive attempts at impossible subgoals by managing not only visited but also unattained subgoals. For this, we can also consider a random sampling over the entire goal space (noticing that this random graph now contains unattained ones and thus differs from SoRB used in the previous works).  However, thanks to the regularity of grid, the grid-based one is more efficient than the random one in terms of the number of subgoals covering the goal space. Indeed, in our additional experiment (Figure 8), the grid-based management (without adaptive refinement) outperforms the random graph. (See Section 5.5 for a comprehensive discussion.) Besides this, the regularity of grid inherently provides the ease of implementing the breadth-first exploration upon adaptive refinement, whereas it is somewhat non-trivial to devise a mechanism of adaptive refinement on the random graph due to the irregular dense, although it seems not impossible.
> >
> > We appreciate the opportunity to further clarify these points and invite any additional feedback or questions from the reviewer.
> > ___
> > **Q9. [Question 6] During planning, you always select the shortest path to the goal in the graph. Does it mean that the agent is committed to this rigid grid-like movement, even if it could move diagonally? Do you see a way to adjust the structure of the graph to eventually allow for near-optimal trajectories? At least, trajectories without considerable inefficiencies?**
> >
> > **A9.**
> > We appreciate your question on the choice of grid pattern. We have used a sparse grid pattern with no diagonal edges just for ease of presentation. Your concern about the sub-optimal planning on the sparse grid can be effortlessly yet effectively closed by using a dense grid pattern with diagonal edges (e.g., $\boxtimes$ in 2D), where one can prove that the worst-case sub-optimality ratio is improved from $\frac{\sqrt{2}}{2} \approx 0.707$ to $\frac{\sqrt{5}}{1+\sqrt{2}} \approx 0.926$ on 2D Euclidean goal space. We will add an experiment with the dense grid pattern if time allows.
> > ___
> > **Q10. [Question 7] It seems to me that this approach is useful if the goal space is low-dimensional. For instance, in 2 dimensions, like ant-maze, it is pretty efficient. However, if we consider any more complex spaces, e.g. ant-maze with the goal space equal to the observation space (which is quite natural in many scenarios), will it still work?**
> >
> > **A10.** We note that graph-based RL algorithms share the reviewer’s concern about the scalability in high-dimensional goal spaces. However, this concern can be addressed by the proposed adaptive refinement, starting from a sufficiently sparse grid and selectively densifying the grid. The grid also provides the aforementioned efficiency from the regularity. In summary, this strategic combination of adaptive refinement and grid structure is designed to offer a scalable solution to high-dimensional goal spaces, as well as the efficiency gained from the regularity of the grid.

---

> > > ### Author Response · Authors · 2023-11-19
> > >
> > > **Q11. [Question 8] If the goal space is reduced, then the reachability of adjacent subgoals depends on the state in which the agent reaches each point, e.g. the exact position of its limbs. In particular, it may depend on the path that it took. How do you handle that?**
> > >
> > > **A11.**
> > > This is indeed an insightful comment. In our understanding, in the environments we consider, the reachability of subgoals depends on the agent’s state. For example, in AntMaze, the agent is often flipped over and cannot recover from this state. However, BEAG seems not to suffer severe issues from noisy estimations of reachability thanks to the roll-out phase, reducing such elementary mistakes. Although our main focus is improving the graph planner (high-level policy) in graph-based RL (implicitly presuming a well-trained low-level policy), it is also interesting to jointly study both the high-level and low-level policies. In fact, we have been extending our work in this direction.

---

> ### Comment · Reviewer_SMnh · 2023-11-21
>
> Thank you for the answer. I acknowledge the clarifications by increasing my rating.

---

### Official Review · Reviewer_z2kz · 2023-10-26

**Soundness:** 2 fair
**Presentation:** 2 fair
**Contribution:** 2 fair
**Rating:** 5
**Confidence:** 4

**Summary:**

This paper proposes a method which builds grid-level graphs for better sample efficiency in subgoal HRL. When building the graph, instead of estimating weights from the Q-function. The authors set edge weights using temporal distance based on the number of successful node visits. The method is also adaptive, which means the graph can fit different grained levels while planning in complex environments.

**Strengths:**

This work has a good motivation: to address the drawbacks of current methods in subgoal-finding process. Some key issues, such as the novelty of subgoals, the accessibility of these subgoals, and sample efficiency, are considered.

The design of experiments clearly shows the sample efficiency of BERG over DHRL.

**Weaknesses:**

1. Experiments are not enough. The authors only compared their work with DHRL, while there exists many other graph-based HRL methods that are worth to compare with, such as HIRO, HAC, ... The effect of threshold hyperparameters are not shown. Also, the ablation study can focus on more aspects of BERG.

2. The method is a minor upgrade of existing Graph-based methods. Although pure empirical evidence can show the effectiveness of the method, I am expecting to see some more theoretical analysis on why it may work.

**Questions:**

1. Could the authors try some other baseline HRL benchmarks to further show the advantage of the proposed method?

2. I would like to see more experiments on failure condition and count thresholds.

---

> ### Author Response · Authors · 2023-11-19
>
> Dear Reviewer z2kz
>
> We would like to express our sincere gratitude for your valuable comments and insightful feedback. Your comment has greatly enriched our work and provided us with valuable perspectives. We address your comments one by one in what follows.
> ___
>
> **Q1 [Weakness 1] Experiments are not enough. The authors only compared their work with DHRL, while there exists many other graph-based HRL methods that are worth to compare with, such as HIRO, HAC, … . [Question 1] Could the authors try some other baseline HRL benchmarks to further show the advantage of the proposed method?**
>
> **A1.**
> As requested, we have expanded our evaluation by incorporating additional baselines, namely PIG [1], HIGL [2], and HIRO [3], alongside DHRL [4]. BEAG outperforms all the baselines, not only graph-based RL approaches (DHRL, PIG, HIGL) but also the other hierarchical RL one (HIRO) while DHRL is the most competitive. We note that our evaluation fairly supports the superiority of BEAG since our choice of benchmarks is standard and comprehensive, aligning with those used by other works to justify their proposals. The observed superiority of BEAG stems from its unique ability to systematically identify and avoid impossible subgoals on the adaptive grid, a factor overlooked by other baselines. For a detailed comparison, please refer to Figure 4 and Section 5.2 in the revised draft, where we present a more comprehensive analysis of the results. We are confident that this expanded comparison offers a clearer understanding of BEAG's strengths, as well as addresses your concerns.
>
>
> [1] Junsu Kim, Younggyo Seo, Sungsoo Ahn, Kyunghwan Son, and Jinwoo Shin. Imitating graph-based planning with goal-conditioned policies. ICLR 2023.
>
> [2] Junsu Kim, Younggyo Seo, and Jinwoo Shin. Landmark-guided subgoal generation in hierarchical
> reinforcement learning. Advances in Neural Information Processing Systems, 34:28336–28349, 2021.
>
> [3] ​​Ofir Nachum, Shixiang Shane Gu, Honglak Lee, and Sergey Levine. Data-efficient hierarchical reinforcement learning. Advances in neural information processing systems, 31, 2018.
>
> [4] Seungjae Lee, Jigang Kim, Inkyu Jang, and H Jin Kim. Dhrl: A graph-based approach for long-
> horizon and sparse hierarchical reinforcement learning.Advances in Neural Information Process-
> ing Systems, 35:13668–13678, 2022.
> ___
> **Q2. [Weakness2] The method is a minor upgrade of existing Graph-based methods. Although pure empirical evidence can show the effectiveness of the method, I am expecting to see some more theoretical analysis on why it may work.**
>
> **A2.** Thank you for the constructive comments. One of our contributions is to address the importance of avoiding consecutive attempts at impossible subgoals, which the literature of graph-based RL (including the most competitive baseline, DHRL) have been overlooked. In addition, we report an additional experiment and discussion, demonstrating a clear advantage of BEAG over DHRL in Figure 5 and Section 5.3 of the revised draft. This result underscores the efficacy of our main idea, the breadth-first exploration.
>
> As you may understand, it is prohibitively challenging to theoretically analyze deep RL. Instead, in response to your request for a theoretical analysis of BEAG, we can provide a guarantee on the grid-based “planner”. Specifically, we can show that given 2D Euclidean goal space, the grid pattern of $\square$ has the worst-case sub-optimality ratio $\frac{\sqrt{2}}{2} \approx 0.707$, and a denser pattern of $\boxtimes$ improves it to $\frac{\sqrt{5}}{1+\sqrt{2}} \approx 0.926$ where ratio 1 implies the optimality. Regarding this, we will include a formal statement and proof as soon as possible.
>
> We appreciate your comment as it provides the opportunity to clarify our contributions and drive an interesting theoretical analysis.
> ___
> **Q3. [Weakness1] The effect of threshold hyperparameters are not shown. Also, the ablation study can focus on more aspects of BEAG. [Question 2] I would like to see more experiments on failure conditions and count thresholds.**
>
> **A3.**
> We appreciate your valuable suggestions on additional experiments. As suggested, in Figure 7, we conducted ablation studies on BEAG with different values of ($\tau_t$: [50, 100, 150, 200]) and ($\tau_n$: [1, 3, 5]).  As expected, both $\tau_t$ and $\tau_n$ (determining the thresholds to avoid subgoals) need to be set at minimal values to reduce attempts at impossible subgoals. However, exceedingly small values for $\tau_t$ and $\tau_n$ may accidentally identify attainable subgoals as impossible ones and possibly degenerate performances in early phases, as depicted in Figure 7 (d) and (e). This observation provides a useful guideline for the hyperparameter choice of our method. We hope that these additional experiments address the reviewer's concern.

---

> ### Comment · Reviewer_z2kz · 2023-11-21
>
> Thank you for the clarification and additional experiments. They help in a better understanding of the paper. I adjusted my score based on the revised version.

---

### Official Review · Reviewer_fvhd · 2023-10-30

**Soundness:** 2 fair
**Presentation:** 1 poor
**Contribution:** 2 fair
**Rating:** 5
**Confidence:** 4

**Summary:**

This paper investigates the issue of inefficient exploration of graph-based methods used in goal-conditioned reinforcement learning. The authors claim that the existing research tends to record achieved goals but overlook the unattained goals, which makes the algorithms struggle to get rid of repeatedly exploring these goals and causing waste. The main idea proposed in this paper is to leverage breadth-first exploration and multi-scale graph-construction to manage achieved and unattained goals.

**Strengths:**

- The paper concentrates on an important and challenging problem of goal-exploration in GRL, which has not been well studied in the literature.
- At least from the lens of writing, the authors propose a simple yet effective solution incorporating a forward-looking perspective and a proxy model for value estimation.
- This paper is easy to follow.

**Weaknesses:**

- As admitted by the authors, the proposed method heavily relies on the task structure (distance-based, discrete) to apply the graph construction to record the achieved and unattained goals.
- The comparative baselines are limited on some environments. Only DHRL is involved. Since RL algorithms often perform high-performance variance in different environments/tasks, I think it is necessary to make comparisons with more baselines. And also, I think it would be better if the authors make comparisons on different environments.
- The authors did not provide clear explanations or intuitions for some of the design choices, such as why we use Euler-distance instead of others as the $w_{i,j}$.
- The illustration in Algorithm tables is unclear (e.g., some notations lack explanation), so it is hard to and even cannot follow that.
- It is unclear how the edge weights work for the proposed method.

**Questions:**

- Why does the comparison in Figures 4-6 not include the other baselines introduced in the Appendix?
- Why does the coverage in Figure 7 decrease?
- Why does the success rate in Figure 4(d) higher than 1?
- Why does the learning curve in Figure 8(a) not start from 0 environment step?

---

> ### Author Response · Authors · 2023-11-19
>
> Dear Reviewer fvhd
>
> We would like to express our sincere gratitude for your valuable comments and insightful feedback. Your comment has greatly enriched our work and provided us with valuable perspectives. We address your comments one by one in what follows.
> ___
>
> **Q1. [Weakness 1]The proposed method heavily relies on the task structure (distance-based, discrete) to apply the graph construction to record the achieved and unattained goals.**
>
> **A1.**
> We appreciate the opportunity to clarify the applicability of our approach. The proposed BEAG is designed to be versatile and can be applied to a broad range of environments where other graph-based RL frameworks are applicable. In essence, the main goal of studying graph-based RL is to develop a framework that fully leverages the distance-based goal structure, a characteristic shared by various practical systems such as navigation and robot-arm manipulation.
>
> Given such a goal structure, BEAG not only applies but also offers distinct advantages, which are twofold: (i) recording achieved and unattained subgoals to avoid consecutive attempts at impossible subgoals; and (ii) selectively densifying the grid to adapt to diverse environments.
>
> Notably, an additional experiment (Figure 8) demonstrates the effectiveness of recording achieved and unattained subgoals, even in a variant of BEAG using a random graph over the entire goal space instead of a grid, while previous works are prone to obsessively attempting impossible subgoals as they rely on SoRB, subsampling subgoals only from a set of visited subgoals. In addition to this, BEAG using grids shows a clear benefit, compared to the variant with random graphs of irregularly distributed subgoals. We refer the reviewer to Section 5.5 for a comprehensive discussion.
>
> We hope this clarification addresses concerns about the reliance on task structure and underscores the versatility and advantages of our approach across various goal-oriented environments.
> ___
> **Q2. [Weakness 2] The comparative baselines are limited on some environments. Only DHRL is involved. Since RL algorithms often perform high-performance variance in different environments/tasks, I think it is necessary to make comparisons with more baselines. And also, I think it would be better if the authors make comparisons on different environments. [Question 2.] Why does the comparison in Figures 4-6 not include the other baselines introduced in the Appendix?**
>
> **A2.** As requested, we have expanded our evaluation by incorporating additional baselines, namely PIG [1], HIGL [2], and HIRO [3], alongside DHRL [4]. BEAG outperforms all the baselines, not only graph-based RL approaches (DHRL, PIG, HIGL) but also the other hierarchical RL one (HIRO) while DHRL is the most competitive. We note that our evaluation fairly supports the superiority of BEAG since our choice of benchmarks is standard and comprehensive, aligning with those used by other works to justify their proposals. The observed superiority of BEAG stems from its unique ability to systematically identify and avoid impossible subgoals on the adaptive grid, a factor overlooked by other baselines. For a detailed comparison, please refer to Figure 4 and Section 5.2 in the revised draft, where we present a more comprehensive analysis of the results. We are confident that this expanded comparison offers a clearer understanding of BEAG's strengths, as well as addresses your concerns.
>
>
> [1] Junsu Kim, Younggyo Seo, Sungsoo Ahn, Kyunghwan Son, and Jinwoo Shin. Imitating graph-based planning with goal-conditioned policies. ICLR 2023.
>
> [2] Junsu Kim, Younggyo Seo, and Jinwoo Shin. Landmark-guided subgoal generation in hierarchical
> reinforcement learning. Advances in Neural Information Processing Systems, 34:28336–28349, 2021.
>
> [3] ​​Ofir Nachum, Shixiang Shane Gu, Honglak Lee, and Sergey Levine. Data-efficient hierarchical reinforcement learning. Advances in neural information processing systems, 31, 2018.
>
> [4] Seungjae Lee, Jigang Kim, Inkyu Jang, and H Jin Kim. Dhrl: A graph-based approach for long-
> horizon and sparse hierarchical reinforcement learning.Advances in Neural Information Process-
> ing Systems, 35:13668–13678, 2022.

---

> ### Author Response · Authors · 2023-11-19
>
> **Q3. [Weakness 3] The authors did not provide clear explanations or intuitions for some of the design choices, such as why we use Euler-distance instead of others as the $w_{i,j}$**
>
> **A3.**
> Thank you for highlighting the need for clarification in our method. In response to your request, we have added an explanation in Section 4.1  to clarify why we opted for Euclidean distance as the weights. We bring it for your convenience:
>
> >We assume that the cost of moving between two subgoals at an equal distance is similar, and thus, we use the Euclidean distance as the weights of the edges. To generalize to diverse environments, if the aforementioned assumption does not hold, the distance measure introduced in Equation 1 can be employed as the weights of the edges. However, it is noteworthy that the Q-function may exhibit instability in predictions for untried transitions, potentially hindering exploration.
>
> Furthermore, our method can adopt alternative edge weights. We are currently conducting an additional experiment using the value difference (Equation 1) for edge weight. We commit to promptly incorporating these results, while we anticipate observing nearly identical advantages of the proposed method. We thank the reviewer for the comment, providing this opportunity to explore other design options and to enhance the transparency of our design choices.
> ___
> **Q4. [Weakness 4] The illustration in Algorithm tables is unclear (e.g., some notations lack explanation), so it is hard to and even cannot follow that.  [Weakness 5] It is unclear how the edge weights work for the proposed method**
>
> **A4.**
> Following your comments, we revised the algorithm in Appendix A and Section 4 for a detailed explanation of our method.
> - change the notation of graph $\mathcal{G}$ into $\mathcal{H}$ for clarification
> - change the notation $a_j, s_j$ into $n_j^a, n_j^s$ in equation (2) for clarification and align them in the algorithm.
> - add the simple explanation of notation in the algorithm.
> - add the algorithm for training and graph construction.
> - add the training details (e.g. initial random stage).
>
> According to your comment on the description of the edge weights in BEAG, we have revised Section 4.1. For the sake of clarity, in what follows, we elaborate on how the edge weights work for BEAG.
> Each edge weight is uniformly initialized with an interval $n$. During an episode, if an assigned subgoal remains unachieved for $\tau_t$ steps, the edges connected to the unattained subgoal in that episode are disconnected by assigning infinite weights to encourage the exploration of new paths.
> If such failures have been continued in the previous $\tau_n$ episodes, the edges will remain disconnected (while the refinement procedure can revive a part of them). Otherwise, they are reconnected in the next episode.
> ___
> **Q5. [Question 2] Why does the coverage in Figure 7 decrease?**
>
> **A5.**
> We note that Figure 7 in the submitted paper has been removed in the revised one. The decrement is just temporal and the coverage is recovered in a few epochs. We present the raw data for Figure 7 in the below table. We hope these have addressed your concern.
>
> |Env steps| 129K | 219K | 309K | 399K | 489K | 579K | 669K |
> | :---: | :---: | :---: | :---: | :---: | :---: | :---: | :---: |
> |Coverage| 0.154 | 0.696 | 0.821 | 0.868 | 0.775 | 0.882 | 0.954 |
> ___
> **Q6. [Question 3] Why is the success rate in Figure 4(d) higher than 1?**
>
> **A6.** We note that Figure 4(d) in the submitted paper has been replaced with Figure 4(e) in the revised one. The shaded region in our plots visualizes the standard deviation centered around the average. Hence, it is possible to observe the shaded region of the success rate curve higher than 1, while there are no sample instances exceeding 1.
> ___
> **Q7. [Question 4] Why does the learning curve in Figure 8(a) not start from 0 environment step?**
>
> **A7.**
> In response to your question, we report the curve starting from the 0 environment step. The revised figure can be found in Figure 4(a) in the revised draft (while it was Figure 8(a) in the submitted paper). BEAG begins with a roll-out phase where the agent proceeds with actions entirely at random without training. This is to obtain a certain size of replay buffer and is also employed in DHRL. In our plots, the success rates of BEAG and DHRL are set to zero over the roll-out phase.

---

> > ### Comment · Reviewer_fvhd · 2023-11-22
> > **Thanks for your response**
> >
> > I thank the authors for their detailed response, and their efforts resolve most of my doubts. However, I still have a concern about Figure 4. As illustrated, the success rate is higher than 1 or lower than zero. I know the shaded region indicates a standard deviation, but this plot surely relies on what your original data is. Thus, I want to know how the authors calculate this measurement, it would be better in formulation.

---

> ### Author Response · Authors · 2023-11-23
>
> Thank you for the positive response. To adderss your remaining concern on our presentation of success rate curves (which should be bounded in [0,1], *instance-wisely*), we have made the following revisions:
> * We have included plots of all the raw data, instance-wisely, in Appendix B.
> * To reduce the potential misunderstanding, we have modified all the figures plotting success rate curves (Figure 4, 5, 6, 7, 8), where we clip the values of (average+-1 standard deviation) of success rates inbetween [0,1]. Apprantely, this cliped presentation is standard in previous works, e.g., HIGL and PIG. Additionally, we would like to emphasize that the computation of the success rate employs the same formula as used in these previous works
>
> We believe these additional plots and modifications will address any concerns or uncertainties you may have had. Please let us know if further clarification or adjustments are needed.

---

### Official Review · Reviewer_qmXA · 2023-11-01

**Soundness:** 3 good
**Presentation:** 3 good
**Contribution:** 2 fair
**Rating:** 6
**Confidence:** 4

**Summary:**

The paper presents an approach for constructing a graph to augment current graph-based planners for goal-conditioned RL, specifically, using a grid that matches the dimension of the goal space and running a shortest path algorithm on the grid graph. In addition, a local adaptive grid refinement and goal shifting are proposed to appropriately choose grid interval and explore unattempted nodes, respectively. Experiments were performed in a number of different tasks in MuJoCo environments and results were compared with state-of-the-art approaches.

**Strengths:**

- The paper addresses a practical subproblem present in current graph-based planners for goal-conditioned RL, that is of wasting attempts to unattainable subgoals.

- The proposed approach is technically sound with the use of a grid and running of shortest path algorithm, and the addition of the grid refinement and goal shifting.

- The experiments include a comprehensive analysis with comparison with other state-of-the-art methods included in the appendix, as well as ablation studies, showing positive results.

- The presentation is clear with a logical structure in introducing the different concepts.

**Weaknesses:**

- while grid is a way to have a systematic representation of the space, there are other approaches in planning that are present in the literature, including sampling-based approaches that can address similar problems. It is worth discussing the rationale behind choosing grid, as that is not necessarily the primary choice for high-dimensional spaces (which the paper mentions to leave it for the future).

- As Reacher3D is the scenario where results are fairly close, goals could be set at different locations to report the coverage too, unless there is a specific reason not to have different goals also for Reacher3D.

Some minor comments on presentation:
- "embarassing" can be substituted with "a very low performance" to be more formal and technical, rather than introducing a subjective judgement.
- "we propose graph planning method dealing with the above problem in Section 4" -> "we propose graph planning method dealing with the above problem in Section 4."
- please ensure that all symbols are introduced, i.e., before Eq. (2), worth mentioning " ... edge weights w_{i,j} for nodes v_i, v_j as follows". Also the symbol for the nodes set should be introduced, when the graph is introduced.
- "The coverage is the averaged success rate of uniformly sampled goals, in which we sample 10 goals per unit-size cell over the
entire goal space to make the sampling more uniform. The coverage represents the average success rate for the entire goal" The two sentences are including redundant information, so they could be merged.
- "Bottleneck-maze where challenging for BEAG" -> "Bottleneck-maze which was challenging for BEAG"
- in Fig. 8, worth changing the max value on the y axis, so that the success rate is completely visible for the bottom row.

**Questions:**

Please  see the first two points included in the Weaknesses box.

---

> ### Author Response · Authors · 2023-11-19
>
> Dear Reviewer qmXA
>
> We would like to express our sincere gratitude for your valuable comments and insightful feedback. Your comment has greatly enriched our work and provided us with valuable perspectives. We address your comments one by one in what follows.
> ___
>
> **Q1. [Weakness 1] While grid is a way to have a systematic representation of the space, there are other approaches in planning that are present in the literature, including sampling-based approaches that can address similar problems. It is worth discussing the rationale behind choosing grid, as that is not necessarily the primary choice for high-dimensional spaces (which the paper mentions to leave it for the future)**
>
> **A1.**
> Thanks for the suggestion. As you mentioned, it is crucial to manage subgoals including not only visited but also unattained ones, in order for avoiding successive attempts to impossible subgoals. To support such subgoal managements, one can also consider a random sampling over the entire goal space (noticing that this random graph now contains unattained ones and thus differs from SoRB used in the previous works). However, thanks to the regularity of grid, the grid-based one is more efficient than the random one in terms of the number of subgoals covering the goal space. Indeed, in our additional experiment (Figure 8), the grid-based management (without adaptive refinement) outperforms the random-sampling. (See Section 5.5 for a comprehensive discussion.) Besides this, the regularity of grid inherently provides the ease of implementing the breadth-first exploration upon adaptive refinement, whereas it is somewhat non-trivial to devise a mechanism of adaptive refinement on the random graph due to the irregular dense, although it seems not impossible.
>
> We note that graph-based RL algorithms share the reviewer’s concern about the scalability in high-dimensional goal spaces. However, this concern can be addressed by the proposed adaptive refinement, starting from a sufficiently sparse grid and selectively densifying the grid. The grid also provides the aforementioned efficiency from the regularity. In summary, this strategic combination of adaptive refinement and grid structure is designed to offer a scalable solution to high-dimensional goal spaces, as well as the efficiency gained from the regularity of the grid.
>
> We appreciate the opportunity to further clarify these points and invite any additional feedback or questions from the reviewer.
> ___
>
> **Q2. [Weakness2] As Reacher3D is the scenario where results are fairly close, goals could be set at different locations to report the coverage too, unless there is a specific reason not to have different goals also for Reacher3D.**
>
> **A2.** We will provide the requested plot comparing the average coverage in Reacher3D, where we believe the main message drawn from the success rate is not different to that from the average coverage.
> ___
> **Q3. Minor comments on the presentation**
>
> **A3.** We appreciate your valuable feedback on the presentation of our work. Specifically, we acknowledge the need for improvement regarding the provocative expression used to describe the performances of the baselines that were initially excluded. Accordingly, we have revised the manuscript to include a clear and comprehensive presentation of results, encompassing the complete set of baselines. We extend our sincere apologies for any unintended tone in the initial description. We have diligently incorporated all of your suggestions into the revised draft.

---

> > ### Comment · Reviewer_qmXA · 2023-11-21
> >
> > Thanks for the response and for updating the paper. Overall the paper appears improved. One note: there are sampling-based approaches that includes bias sampling towards the boundaries of the obstacles, thus improving a vanilla random sampling for bottleneck scenarios.

---

> > > ### Author Response · Authors · 2023-11-23
> > >
> > > We sincerely appreciate your constructive comments, which have greatly contributed to the enhancement of our paper. As you noted, we agree that the random sampling we used in the additional experiment (Figure 8) can be further improved by adopting a variant of our adaptive refinement for the random graph. However, we believe that there should be the advantages of the grid-based method thanks to its regularity, as discussed in our previous response. To further clarify the rationale behind choosing the grid-based method in BEAG, we kindly request your guidance in directing us to relevant papers that propose the sampling-based approach you mentioned.

---

### Author Response · Authors · 2023-11-19
**General response for all reviewers**

Dear reviewers and AC,

Thank you for all of the valuable feedback and suggestions on our research. Your suggestions have been instrumental in enhancing our work.

In response to the questions and concerns, we have carefully revised and improved the draft with the following additional experiments and discussions:

* Clarification of our notation
* Clarification about the description and pseudo code of the proposed method, BEAG  (Section 4, Appendix A)
* Additional baselines (PIG, HIGL, HIRO) for experiments (Section 5.2)
* Additional experiments, highlighting clear gains of BEAG (fixed goal environment) (Section 5.3)
* Additional ablation studies, providing an insight into the hyperparameter choice of BEAG  (random graph, failure condition, and count) (Section 5.5)
* Specifying the hyperparameters for baselines (Appendix B; our implementation code)

~~Although most of the additional experiments have been conducted and reported in response to the reviewers’ comments and questions, we notice that some baselines are yet to be included in experiments (Bottleneck-maze: HIGL, HIRO, Complex-maze: HIRO, $\pi$-maze: HIRO). However, we commit to completing the experiments and updating the results with the missing baselines before the end of the rebuttal period.~~

We have updated the results with the missing baselines. (Figure 4)

These updates are temporarily highlighted in "$\textcolor{magenta}{magenta}$" for your convenience to check.

Thank you,

Authors.

---

### Meta-Review · Area_Chair_SiZP · 2023-12-11

**Metareview:**

The paper proposes a solution to manage exploration in goal-based RL by reducing wasted attempts to achieve an unachievable subgoal. The proposed solution leverages a grid representation of the goal space and tracks statistics of the agent’s capability to achieve each grid subgoal. Graph-based planning is used to identify achievable paths through the goal space. The reviewers agree that the solution seems plausible for environments where a grid-based representation is feasible and Euclidean distances between subgoals are appropriate. They also note that this work presents a minor upgrade of existing methods. However, the paper does not present detailed experiments or theories to justify that it will behave differently from the other methods. There are also some claims in the paper, e.g., the algorithm produces lower variance in performance during training, that are not supported (only four trials), and statistical uncertainty and variations due to hyperparameter tuning are ignored. As such, I do not recommend this paper for acceptance in its current state.

**Justification For Why Not Higher Score:**

This paper should not be accepted because it justifies that its method behaves as intended with only performance measures. In addition to these measures not being able to show anything but performance, the experiments that generate them do not contain enough trials to account for uncertainty sufficiently.

**Justification For Why Not Lower Score:**

N/A

---

### Decision · Program_Chairs · 2024-01-16

Reject